# Enantiocontrolled Preparation of ϒ-Substituted Cyclohexenones: Synthesis and Kinase Activity Assays of Cyclopropyl-Fused Cyclohexane Nucleosides

**DOI:** 10.3390/ijms23179704

**Published:** 2022-08-26

**Authors:** Sergio Jurado, Beatriz Domínguez-Pérez, Ona Illa, Jan Balzarini, Félix Busqué, Ramon Alibés

**Affiliations:** 1Departament de Química, Universitat Autònoma de Barcelona, Bellaterra, 08193 Barcelona, Spain; 2Laboratory of Virology and Chemotherapy, Rega Institute, Katholieke Universiteit Leuven, B-3000 Leuven, Belgium

**Keywords:** asymmetric synthesis, carbocyclic nucleosides, HSV-1 thymidine kinase, enzymatic assays

## Abstract

The enantioselective preparation of the two isomers of 4-hydroxy-2-cyclohexanone derivatives **1a**,**b** was achieved, starting from a common cyclohexenone, through asymmetric transfer hydrogenation (ATH) reactions using bifunctional ruthenium catalysts. From these versatile intermediates, a stereoselective route to a cytosine analogue built on a bicyclo [4.1.0]heptane scaffold is described. Nucleoside kinase activity assays with this cyclopropyl-fused cyclohexane nucleoside, together with other related nucleosides (**2a**–**e**), were performed, showing that thymine- and guanine- containing compounds have affinity for herpes simplex virus Type 1 (HSV-1) thymidine kinase (TK) but not for human cytosolic TK-1, thus pointing to their selectivity for herpetic TKs but not cellular TKs.

## 1. Introduction

Optically active ϒ-substituted cycloalkenones are compounds of synthetic importance that are used as precursors in the synthesis of natural products and pharmaceutically active molecules [1,2,3,4]. In particular, both enantiomers of 4-hydroxy-2-cyclohexenone **1a** and their *O*-protected derivatives, such as **1b**,**c**, have been extensively used in organic synthesis (Figure 1) [5,6,7]. These chiral cycloalkenones have been prepared by different methodologies that involve enzymatic transformations [8], kinetic resolutions [9], chiral auxiliaries [10], asymmetric catalysis [11], or chiral pool compounds [12,13]. However, most of these approaches do not provide access to both enantiomers. Therefore, the development of efficient routes to such relevant building blocks is of considerable interest.

In a previous work, we reported the synthesis of the nucleoside analogues (NAs) **2a**–**d** built on a bicyclo [4.1.0]heptane scaffold, starting from cyclohexenone **3** bearing a dihydrobenzoin moiety as the chiral auxiliary [14]. These NAs were designed as potential antiherpetic agents by a molecular modeling study on the HSV-1 TK active site that is involved in the first phosphorylation step of the activation process. However, none of the compounds showed significant antiviral activity at subtoxic concentrations (~250 μM). The lack of activity indicated that these functionalized NAs might, eventually, fail to reach the HSV DNA polymerase interaction step. Therefore, further studies were needed to identify the molecular basis of the antiherpetic inactivity of the prepared compounds; in particular, single enzymatic assays were required to test the suitability of these compounds to pass the usual rate-limiting first phosphorylation step [15].

In order to perform these enzymatic assays, to complete the 5′-hydroxymethylbicyclo [4.1.0]heptanyl family **2** by preparing the cytosine analogue **2e**, and considering our work on the synthesis of bioactive products bearing a cyclohexane unit [16,17,18], general and easy access to both enantiomers of **1a** and **1b** was required. We envisaged that the asymmetric transfer hydrogenation (ATH) on 1,4-cyclohexenedione monoethylene ketal **4** could be applied to prepare such chiral cyclohexenones [19]. ATH with bifunctional ruthenium catalysts has become one of the most practical and versatile tools for accessing enantiomerically enriched alcohols in organic synthesis, due to its excellent selectivity, practical simplicity, and wide substrate scope [20,21]. In this article, we disclose a concise entry to both enantiomers of ϒ-hydroxycyclohexenone by ATH via catalyst control, their use as precursors in the preparation of the conformationally restricted cytosine analogue **2e**, and the results of the enzymatic affinity of compounds **2a**–**e** on HSV-1 thymidine kinase.

## 2. Results and Discussion

The synthesis of both enantiomers of **1a** and **1b** (Figure 1) started with enone **4**, which was easily prepared on a multigram scale from the commercially available 1,4-cyclohexanedione monoethylene acetal in 88% yield, through a two-step optimized dehydrogenation protocol that involved the formation of the corresponding silyl enol ether followed by oxidation with IBX·MPO complex [17]. The asymmetric reduction of enone **4** was first carried out in a biphasic medium (CH_2_Cl_2_/H_2_O 1:1), using (*R,R*)-Noyori-I, (*R,R*)-**5** as catalyst, TBAC as a phase-transfer agent, and HCOONH_4_ as the hydrogen source [22,23,24]. Under these experimental conditions, allylic alcohol (*R*)-**6** was obtained, along with its totally hydrogenated derivative **7** in a 6:1 ratio and 67% yield. The reduction also resulted in the removal of the ketal protecting group, providing variable amounts of **1a**. However, when the reaction was performed using HCOONa [25], the desired allylic alcohol (*R*)-**6** was isolated in 79% yield. Next, deprotection of the carbonyl group was achieved by treatment with montmorillonite K-10 in dichloromethane at room temperature, affording the volatile 4-hydroxy-2-cyclohexenone (*R*)-**1a** in 64% yield and 92% ee (determined by CHPLC). Protection of the alcohol **4** as a silyl ether with imidazole and TBSCl in dichloromethane and subsequent hydrolysis of the ketal using montmorillonite K-10 led to (*R*)-**1b** in 87% yield for the two steps.

In a similar way, the asymmetric reduction was performed with the enantiomeric (*S,S*)-**5** catalyst, providing allylic alcohol (*S*)-**6** in 78% yield, along with traces of the totally hydrogenated derivative. Hydrolysis of the ketal delivered (*S*)-**1a** in 62% yield and 92% ee (determined by CHPLC), while silyl ether protection and ketal cleavage afforded (*S*)-**1b** in 88% yield. Thus, we established a practical enantioselective procedure to prepare both enantiomers of 4-hydroxy-2-cyclohexenone (*R*)- and (*S*)-**1a** in four steps, ca. 44% overall yield and 92% ee, and their *O*-silyl derivatives, (*R*)- and (*S*)-**1b,** in five steps and 60% overall yield, which compared with other published methodologies [5,6,7,8,9,10,11,12,13]. Both enantiomers of the product could be obtained from the same precursor only by selecting either the (*S*,*S*)- or (*R*,*R*)-**5** catalyst.

The satisfactory results achieved in the ATH step prompted us to examine the scope of the reaction with other cyclohexenones **8**–**10** using the optimized conditions ((*R,R*)-**5**, TBAC and HCOONa in CH_2_Cl_2_/H_2_O 1:1, Table 1). The ATH reaction on 2-cyclohexenone **8** delivered a 2:1 mixture of the corresponding alcohol **11** and the fully hydrogenated cyclohexanol **12** in good yield with an enantiomeric excess of 92% (entry 1). The ATH on cyclohexenone **9** bearing a methyl group at the β position, provided a 10:1 mixture of the corresponding allylic alcohol **13** and its fully hydrogenated derivative **14**, from which the major compound **13** could be isolated by column chromatography (entry 2). The enantioselectivity of the reaction was slightly lower (88% ee). Finally, the ATH reaction with 1-tetralone **10** (entry 3) delivered allylic alcohol **15** in good yield with the highest enantioselectivity (94% ee). Overall, the ATH reaction on **8**–**10** in the presence of (*R,R*)-**5** proved to be highly effective, providing access to the corresponding alcohols as the major products with excellent levels of enantioselectivity. The enantio- and chemoselectivities of the ATH reaction were sensitive to the structure of the cyclohexenones. In cyclohexanone **8**, the formation of the fully hydrogenated compound was detrimental for the yield. The presence of a vinylic methyl group enhances the chemoselectivity of the ATH reaction. These chiral cyclohexanols are useful platforms for the synthesis of more elaborate products [26,27,28]. The ATH reaction has also been studied on cycloalkenones, with five- and seven-membered rings obtaining lower values of enantio- and chemoselectivity (see the Appendix A).

Next, we turned our attention to the use of chiral cyclohexanol *R*-(**6**) as the starting material in the preparation of the cytosine analogue **2e,** following an adaptation of our earlier work [14]. Accordingly, the synthesis of **2e** (Figure 2) started with the cyclopropanation reaction on (*R*)-**6**, which was first attempted using Shi’s carbenoid (CF3COOZn CH2I) [29,30]. Under these conditions, the cyclopropanation took place with concomitant removal of the ketal to deliver the known bicyclic keto alcohol **16** albeit in very low yield (16%). Better results were obtained using Furukawa’s procedure [31,32] (Et_2_Zn and ICH_2_Cl), which, after purification by column chromatography, afforded **16** in 78% yield. This compound was highly volatile; therefore, the next protection step was performed immediately after the cyclopropanation reaction without further purification. To continue with the synthesis, two protecting groups were evaluated. First, protection of the alcohol as a *tert*-butyldiphenylsilyl ether furnished **17** in 22% yield for the two steps. All attempts to increase the yield failed. On the other hand, benzyl protection under standard conditions (BnBr, NaH, THF) did not provide better results, delivering only unidentified decomposition products. After some experimentation, it was found that the *O*-benzyl-protected **18** could be obtained in 62% overall yield over the two steps, using BnBr and Ag_2_O. Next, olefination of the ketone via a Wittig reaction led to the expected alkene **19,** which was rapidly submitted to the next step, as it easily isomerizes to the endocyclic isomer **20**. Accordingly, the subsequent hydroboration (9-BBN) -oxidation (H_2_O_2_) process furnished a chromatographically inseparable 2:1 mixture of diastereomers **21** and **22** in 95% yield. After several purifications by column chromatography, an enriched fraction of the main product was obtained and analyzed by NMR. The anti relative configuration of C-2′ of the main product was determined by a NOESY experiment that showed cross peaks between H-7′endo and H-2′, indicating that the approach of the borane through the syn face was favored, due to the steric hindrance between the alkylborane and the cyclopropane in the four-center transition state that resulted from the approach of the borane to the more accessible anti face. After purification by column chromatography, benzoyl protection, followed by the removal of the benzyl-protecting group, delivered alcohol **23** in 62% yield and its diastereomer **24** in 29% yield. The primary amine **25** required for the nucleobase construction was accomplished in 84% yield, starting from **23** through a Mitsunobu reaction using diphenylphosphoryl azide (DPPA) [33], followed by catalytic hydrogenation and hydrochloride salt formation.

The cytosine nucleoside was prepared through the amination of the corresponding uridine derivative, which was prepared by a two-step protocol that involved the addition of the amine **25** to the isocyanate **26**, followed by acid-mediated cyclization, to furnish the protected uracil nucleoside **27** [34]. Then, a one-pot amination in the presence of TsCl, Et_3_N and *N*-methylpiperidine at 0 °C [35], followed by ammonolysis with 30% NH_4_OH solution in water and, finally, removal of the benzoyl protection (CH_3_NH_2_, 33% in EtOH), afforded, after purification by column chromatography, the desired cytosine nucleoside analogue **2e**.

Compound **2e** was examined for antiherpetic activity (herpes simplex virus-1 (HSV-1; strain KOS) and herpes simplex virus-2 (strain G) in human embryonic lung (HEL) cell cultures. Unfortunately, however, it did not show significant antiviral activity at subtoxic concentrations (~100 μM) (see Appendix A). These results were similar to those obtained for the previously synthesized compounds **2a**–**d**. As previously mentioned, the lack of significant activity against herpes simplex virus by the 5′-hydroxymethylbicyclo [4.1.0]heptanyl NAs may arise from a low affinity (if any) or from substrate activity for the viral-encoded or cellular nucleoside kinases. Therefore, to evaluate this last point, we performed further studies to test the suitability of these compounds to pass the usual rate-limiting first phosphorylation step. Thus, the affinity of the synthesized compounds **2a**–**e** for the cellular and HSV-1 thymidine kinases (TKs) were investigated (Table 2). Based on these results, it can be concluded that compounds **2b** and, to a lesser extent, **2d**, were nicely recognized by herpes simplex virus-TK (IC_50_ = 1.6 ± 0.1 µg/mL and 52 ± 28 µg/mL, respectively), in agreement with our modelling studies [14]. Only compound **2b** showed some marginal affinity for mitochondrial TK-2, in addition to its excellent affinity for HSV-1 TK. The **2b** analogue was also evaluated for substrate activity by HPLC technology, and it was found to be a good substrate for HSV-1 TK with an efficient conversion to the monophosphate metabolite. The cytosine derivative **2e** did not display affinity for HSV-1 TK. It was expected that this cytosine analogue might show affinity for HSV-1 TK, as the natural nucleoside (2-deoxycytidine, 2-dC) is known to be recognized by several kinases, including HSV-1 TK; however, that was apparently not the case. Interestingly, none of the analogues showed significant affinity to human cellular kinases. Therefore, because some of the synthesized analogues were recognized by the virus-encoded nucleoside kinase, the lack of antiherpetic activity could be attributed to the lack of further conversion of the 5′-monophosphate derivative to the 5′-triphosphate or to a low affinity, if any, of the 5′-triphosphate metabolite to the virus-encoded DNA polymerase.

## 3. Materials and Methods

**General Methods.** Commercially available reagents were used as received. Solvents were dried by distillation over the appropriate drying agents. All of the reactions were monitored by analytical thin-layer chromatography (TLC), using silica gel 60 F254 pre-coated aluminum plates (0.25 mm thickness). TLC spots were detected under UV light and/or by charring with a KMnO_4_/KOH aqueous solution or vanillin solution. Flash column chromatography was performed using silica gel (230–400 mesh). ^1^H NMR spectra were recorded using 250 or 400 MHz and were referenced to the residual proton signals of CDCl_3_, 7.26 ppm, and MeOH-*d*_4_, 3.31 ppm. ^13^C [^1^H] NMR spectra were recorded at 90 MHz or 100 MHz and were referenced to the residual ^13^C signal of CDCl_3_, 77.16 ppm, and MeOH-*d*_4_, 49.00 ppm. NMR signals were assigned with the help of COSY, HSQC, HMBC, and NOESY experiments. Melting points were determined on a hot stage and were uncorrected. Optical rotations were measured at 20 ± 2 °C at the sodium D line (589 nm) in a microcell (0.1 dm). Infrared spectra were recorded on a spectrophotometer equipped with a Golden Gate Single Refraction Diamond ATR (attenuated total reflectance) accessory. High-resolution mass spectra were recorded using electrospray ionization (ESI).

**Thymidine Kinase Assay Using [CH_3_-^3^H]dThd as the Natural Substrate** [36]. The activity of recombinant thymidine kinase 1 (TK-1), TK-2, and herpes simplex virus-1 (HSV-1) TK, and the 50% inhibitory concentration of the test compounds, were assayed in a 50-μL reaction mixture containing 50 mM Tris/HCl, pH 8.0, 2.5 mM MgCl_2_, 10 mM dithiothreitol, 0.5 mM CHAPS, 3 mg/mL bovine serum albumin, 2.5 mM ATP, 1 µM [methyl-^3^H]dThd, and enzyme. The samples were incubated at 37 °C for 30 min in the presence or absence of different concentrations (five-fold dilutions) of the test compounds. At this time, the enzyme reaction still proceeded linearly. Aliquots of 45 µL of the reaction mixtures were spotted on Whatman DE-81 filter paper disks (Whatman, Clifton, NJ, USA). The filters were washed three times for 5 min each in 1 mM ammonium formate, once for 1 min in water, and once for 5 min in ethanol. The radioactivity was determined by scintillation counting.

### Synthetic Procedures

**(8*R*)-1,4-Dioxaspiro [4.5]dec-6-en-8-ol [(*R*)-6].** To a stirred solution of enone **4** (104 mg, 0.68 mmol) in a biphasic medium of CH_2_Cl_2_ (2.3 mL) and water (2.3 mL), tetrabutylammonium chloride (57 mg, 0.20 mmol), sodium formate (139 mg, 2.04 mmol), and (*R,R*)-Noyori-I [(*R,R*)-**5**] catalyst (13 mg, 0.020 mmol) were sequentially added at room temperature. The reaction mixture was allowed to stir for 3 h. Then, the phases were separated, and the aqueous layer was extracted with additional CH_2_Cl_2_ (2 × 2 mL) and the organic layers were concentrated under reduced pressure, affording a yellow oil that was purified by column chromatography (diethyl ether) to furnish allylic alcohol (*R*)-**6** (83 mg, 0.53 mmol, 79%): [α]_D_^20^ = +39.0 (*c* 1.2, CHCl_3_) [lit: [37] [α]_D_^20^ = +39.8 (*c* 1.21, CHCl_3_), ee = 93.1%] as a yellowish oil: ^1^H NMR (250 MHz, CDCl_3_) δ 5.94 (ddd, *J*_7,6_ = 10.1 Hz, *J*_7,8_ = 2.8 Hz, *J*_7,9_ = 1.1 Hz, 1H, H-7), 5.61 (dt, *J*_6,7_ = 10.1 Hz, *J*_6,8_ = *J*_6,10_ =1.5 Hz, 1H, H-6), 4.29–4.12 (m, 1H, H-8), 4.05–3.85 (m, 4H, H-2, H-3), 2.17–2.02 (m, 1H, H-9), 2.00–1.89 (m, 1H, H-9), 1.85–1.65 (m, 2H, H-10); ^13^C NMR (90 MHz, CDCl_3_) δ 135.3 (C-7), 129.0 (C-6), 105.2 (C-5), 66.0 (C-8), 64.8/64.6 (C-2, C-3), 31.0/30.6 (C-9, C-10).

**(8*S*)-1,4-Dioxaspiro [4.5]dec-6-en-8-ol [(*S*)-6].** To a stirred solution of enone **4** (76 mg, 0.49 mmol) in a biphasic medium of CH_2_Cl_2_ (1.3 mL) and water (1.3 mL), tetrabutylammonium chloride (45 mg, 0.16 mmol), sodium formate (140 mg, 2.06 mmol), and (*S,S*)-Noyori-I catalyst, [(*S,S*)-**5**], (9.1 mg, 0.015 mmol) were sequentially added at room temperature. The reaction mixture was allowed to stir for 24 h. Then, the phases were separated, and the aqueous layer was extracted with CH_2_Cl_2_ (2 × 10 mL). The organic layers were concentrated under reduced pressure, affording a yellow oil that was purified by column chromatography (diethyl ether) to obtain allylic alcohol (*S*)-**6** (60 mg, 0.38 mmol, 78% yield) as a yellow oil: [α]_D_^20^ = −38.7 (*c* 1.3, CHCl_3_) [lit: [38] [α]_D_^20^ = −40.5 (*c* 1.24, CHCl_3_) ee > 98.4%]; ^1^H NMR (250 MHz, CDCl_3_) δ: 5.93 (ddd, *J*_7,6_ = 10.1 Hz, *J*_7,8_ = 2.8 Hz, *J*_7,9_ = 1.0 Hz, 1H, H-7), 5.60 (dt, *J*_6,7_ = 10.1 Hz, *J*_6,8_ = *J*_6,10_ = 1.4 Hz, 1H, H-6), 4.26–4.16 (m, 1H, H-8), 4.05–3.85 (m, 4H, H-2, H-3), 2.19–2.05 (m, 1H, H-9), 2.04–1.90 (m, 1H, H-9), 1.90–1.68 (m, 2H, H-10).

**(4*R*)-4-Hydroxycyclohex-2-en-1-one [(*R*)-1a]**. Montmorillonite K-10 (832 mg) was added to a solution of (*R*)-**6** (77 mg, 0.49 mmol) in CH_2_Cl_2_ (7.6 mL) and the mixture was stirred at room temperature for 4 h. Then, it was filtered, and the solvent was removed under vacuum to furnish the enone (*R*)-**1a** (34 mg, 0.31 mmol, 64% yield) as a yellowish oil: CHPLC (Daicel Chiralpak IC): 92% ee; [α]_D_^20^ = +90.0 (*c* 0.2, CHCl_3_) [lit: [39] [α]_D_^20^ = +92.3 (*c* 0.7, CHCl_3_) ee > 99%]; ^1^H NMR (250 MHz, CDCl_3_) δ 6.95 (ddd, *J*_3,2_ = 10.2 Hz, *J*_3,4_ = 2.1 Hz, *J*_3,5eq_ = 1.6 Hz, 1H, H-3), 5.98 (ddd, *J*_2,3_ = 10.2 Hz, *J*_2,4_ = 2.1 Hz, *J*_2,6_ = 1.0 Hz, 1H, H-2), 4.58 (ddt, *J*_4,5ax_ = 9.2 Hz, *J*_4,5eq_ = 4.4 Hz, *J*_4,2_ =*J*_4,3_ =2.1 Hz, 1H, H-4), 2.60 (dt, *J*_gem_ = 9.8 Hz, *J*_6eq,5_ = *J*_6eq,4_ = 4.5 Hz, 1H, H-6eq), 2.45–2.28 (m, 2H, H-6ax, H-5eq), 2.10–1.90 (m, 1H, H-5ax); ^13^C NMR (90 MHz, CDCl_3_) δ 199.4 (C-1), 153.6 (C-3), 129.0 (C-2), 66.2 (C-4), 35.4 (C-6), 32.4 (C-5).

**(4*S*)-4-Hydroxycyclohex-2-en-1-one [(*S*)-1a]**. Montmorillonite K-10 (852 mg) was added to a solution of (*S*)-**6** (79 mg, 0.50 mmol) in CH_2_Cl_2_ (7.8 mL) and the mixture was stirred at rt for 4 h. Then, it was filtered and the solvent was removed under vacuum to furnish the known enone (*S*)-**1a** (35 mg, 0.31 mmol, 62% yield) as a yellowish oil: CHPLC (Daicel Chiralpak IC): 92% ee; [α]_D_ ^20^ =−92.3 (c 0.50, CHCl_3_) [lit: [9] [α]_D_^20^= −92.0 (c 0.50, CHCl_3_) ee > 99%]; ^1^H NMR (250 MHz, CDCl_3_) δ 6.93 (ddd, *J*_3,2_ = 10.2 Hz, *J*_3,4_ = 2.4 Hz, *J*_3,5eq_ = 1.6 Hz, 1H, H-3), 5.95 (ddd, *J*_2,3_ = 10.2 Hz, *J*_2,4_ = 2.0 Hz, *J*_2,6_ = 1.0 Hz, 1H, H-2), 4.68–4.46 (m, 1H, H-4), 2.80–2.68 (m, 1H, OH), 2.57 (ddd, *J*_gem_ = 9.8 Hz, *J*_6eq,4_ = 5.0Hz, *J*_6eq,5_ = 4.0 Hz, 1H, H-6eq), 2.47–2.24 (m, 2H, H-6ax, H-5eq), 2.00 (ddd, *J*_gem_ = 12.8 Hz, *J*_5ax,6ax_ = 9.5 Hz, *J*_5ax,4_ = 5.3 Hz, 1H, H-5ax).

**(4*R*)-4-((*tert*-Butyldimethylsilyl)oxy)cyclohex-2-en-1-one [(*R*)-1b].** To a stirred solution of (*R*)-**6** (102 mg, 0.65 mmol) in CH_2_Cl_2_ (0.5 mL), imidazole (56 mg, 0.82 mmol) and a solution of TBSCl (124 mg, 0.82 mmol) in CH_2_Cl_2_ (0.4 mL) were added at room temperature. The mixture was allowed to stir overnight. Then, a saturated aqueous NaHCO_3_ solution (1 mL) was added, the aqueous layer was extracted with CH_2_Cl_2_ (3 × 1 mL), and the organic layers were dried and concentrated under vacuum. The crude was dissolved in CH_2_Cl_2_ (4 mL) and montmorillonite K-10 (513 mg) was added. The reaction mixture was allowed to stir at room temperature for 1 h. Then, it was filtered, and the solvent was removed under vacuum to furnish enone (*R*)-**1b** (128 mg, 0.56 mmol, 87% yield) as a yellowish oil: [α]_D_^20^ = +92.3 (*c* 1.02, CHCl_3_) [lit: [39] [α]_D_^20^ = +97.9 (*c* 1.2, CHCl_3_) ee > 99%]; ^1^H NMR (400 MHz, CDCl_3_) δ 6.82 (ddd, *J*_3,2_ = 10.2 Hz, *J*_3,4_ = 2.1 Hz, *J*_3,5eq_ = 1.6 Hz, 1H, H-3), 5.91 (ddd, *J*_2,3_ = 10.2 Hz, *J*_2,4_ = 2.1 Hz, *J*_2,6eq_ = 1.0 Hz, 1H, H-2), 4.51 (ddt, *J*_4,5ax_ = 9.1 Hz, *J*_4,5eq_ = 4.6 Hz, *J*_4,2_ = *J*_4,3_ = 2.1 Hz, 1H, H-4), 2.56 (dt, *J*_gem_ = 16.8 Hz, *J*_6eq,5ax_ = *J*_6eq,5eq_ = 4.5 Hz, 1H, H-6eq), 2.33 (ddd, *J*_gem_ = 16.8 Hz, *J*_6ax,5ax_ = 12.8 Hz, *J*_6ax,5eq_ = 4.7 Hz, 1H, H-6ax), 2.20 (dqd, *J*_gem_ = 12.8 Hz, *J*_5eq,4_ = *J*_5eq,6eq_ = *J*_5eq,6ax_ = 4.6 Hz, *J*_5eq,3_ = 1.6 Hz, 1H, H-5eq), 1.99 (tdd, *J*_gem_ = *J*_5ax,6ax_ = 12.8 Hz, *J*_5ax,4_ = 9.1 Hz, *J*_5ax,6eq_ = 4.5 Hz, 1H, H-5ax), 0.90 (s, 9H, H-*t*Bu), 0.12 (s, 3H, SiCH_3_), (s, 3H, SiCH_3_).

**(4*S*)-4-((*tert*-Butyldimethylsilyl)oxy)cyclohex-2-en-1-one [(*S*)-1b].** To a stirred solution of (*S*)-**6** (49 mg, 0.32 mmol) in CH_2_Cl_2_ (0.4 mL), imidazole (34 mg, 0.49 mmol) and a solution of TBSCl (74 mg, 0.49 mmol) in CH_2_Cl_2_ (0.2 mL) were added at room temperature. The mixture was allowed to stir overnight. Then, a saturated aqueous NaHCO_3_ solution (1 mL) was added, the aqueous layer was extracted with more CH_2_Cl_2_ (3 × 1 mL), and the organic layers were dried and concentrated under vacuum. The crude was dissolved in CH_2_Cl_2_ (4 mL) and montmorillonite K-10 (294 mg) was added. The reaction mixture was allowed to stir at room temperature for 1 h. Then, it was filtered, and the solvent was removed under reduced pressure to afford enone (*S*)-**1b** (64 mg, 0.28 mmol, 88% yield) as a yellowish oil: [α]_D_^20^= −93.7 (*c* 0.70, CHCl_3_) [lit: [6] [α]_D_^20^ = −97.0 (*c* 1.26, CHCl_3_), ee >99%]; ^1^H NMR (250 MHz, CDCl_3_) δ 6.81 (ddd, *J*_3,2_ = 10.2 Hz, *J*_3,4_ = 2.1 Hz, *J*_3,5eq_ = 1.6 Hz, 1H, H-3), 5.90 (ddd, *J*_2,3_ = 10.2 Hz, *J*_2,4_ = 2.1 Hz, *J*_2,6eq_ = 1.0 Hz, 1H, H-2), 4.50 (ddt, *J*_4,5ax_ = 9.0 Hz, *J*_4,5eq_ = 4.8 Hz, *J*_4,3_ = *J*_4,2_ = 2.1 Hz, 1H, H-4), 2.55 (dt, *J*_gem_ = 16.7 Hz, *J*_6eq.5ax_ = *J*_6eq,5eq_ = 4.8 Hz, 1H, H-6eq), 2.32 (ddd, *J*_gem_ = 16.7 Hz, *J*_6ax,5ax_ = 12.8 Hz, *J*_6ax,5eq_ = 4.8 Hz, 1H, H-6ax), 2.19 (dqd, *J*_gem_ = 12.8 Hz, *J*_5eq,6eq_ = *J*_5eq,4_ = *J*_5eq,6ax_ = 4.8 Hz, *J*_5eq,3_ = 1.6 Hz, 1H, H-5eq), 1.97 (tdd, *J*_gem_ = *J*_5ax,6ax_ = 12.8 Hz, *J*_5ax,4_ = 9.0 Hz, *J*_5ax,6eq_ = 4.8 Hz, 1H, H-5ax), 0.89 (s, 9H, H-*t*Bu), 0.11 (s, 3H, SiCH_3_), 0.10 (s, 3H, SiCH_3_).

**(1*R*)-2-Cyclohexen-1-ol [(*R*)-11].** To a stirred solution of enone **8** (250 µL, 2.59 mmol) in a biphasic medium of CH_2_Cl_2_ (8.6 mL) and water (8.6 mL) and under a nitrogen flux, tetrabutylammonium chloride (222 mg, 0.8 mmol), sodium formate (532 mg, 7.82 mmol), and (*R,R*)-Noyori-I catalyst, [(*R,R*)-**5**] (49.6 mg, 0.08 mmol) were sequentially added at room temperature. The reaction mixture was allowed to stir for 24 h. Then, CH_2_Cl_2_ (10 mL) and water (10 mL) were added, the two phases were separated, and the aqueous layer was extracted with CH_2_Cl_2_ (3 × 5 mL). The volatiles of the organic phase were removed under reduced pressure and the resulting residue was purified by column chromatography (CHCl_3_), affording an inseparable mixture of allylic alcohol (*R*)-**11** and cyclohexanol **12** (210 mg, 2.14 mmol, 82% yield) in a 2:1 ratio as a yellowish oil that was analyzed by NMR spectroscopy and CHPLC (Daicel Chiralpak IC): 92% ee; ^1^H NMR (250 MHz, CDCl_3_) ((*R*)-**11**) δ 5.82 (dtd, *J*_3,2_ = 10.0 Hz, *J*_3,4eq_ = *J*_3,4ax_ = 3.3 Hz, *J*_3,1_ = 1.1 Hz, 1H, H-3), 5.73 (ddt, *J*_2,3_ = 10.0, *J*_2,1_ = 3.4, *J*_2,4_ = *J*_2,6_ = 1.8 Hz, 1H, H-2), 4.42–4.13 (m, 1H, H-1), 2.07–1.89 (m, 1H, H-6), 1.92–1.81 (m, 1H, H-6), 1.79–1.64 (m, 2H, H-4, H-5), 1.64–1.48 (m, 2H, H-4, H-5); (**12**) δ 3.66–3.52 (m, 1H, H-1′), 1.64–1.48 (m, 4H, H-2′, H-6′), 1.38–0.96 (m, 6H, H-3′, H-4′, H-5′).

**(1*R*)-3-Methyl-2-cyclohexen-1-ol [(*R*)-13].** To a stirred solution of enone **9** (510 µL, 495 mg, 4.49 mmol) in a biphasic medium of CH_2_Cl_2_ (14.7 mL) and water (14.7mL) and under a nitrogen flux, tetrabutylammonium chloride (367 mg, 1.32 mmol), sodium formate (899 mg, 13.2 mmol), and (*R,R*)-Noyori-I catalyst [(*R,R*)-**5**] (84 mg, 0.13 mmol) were sequentially added at room temperature. The reaction mixture was allowed to stir for 72 h. Then, CH_2_Cl_2_ (10 mL) and water (10 mL) were added, the two phases were separated, and the aqueous layer was extracted with CH_2_Cl_2_ (3 × 5 mL). The volatiles of the organic phase were removed under reduced pressure and the resulting residue was purified by column chromatography (CHCl_3_), yielding allylic alcohol (*R*)-**13** (327 mg, 2.92 mmol, 65% yield) as a yellowish oil: CHPLC (Daicel Chiralpak IC): 88% ee; [α]_D_^20^ = +89.8 (*c* 0.1, CHCl_3_) [lit: [40] [α]_D_^20^ = +62.4 (*c* 1.0, CHCl_3_), ee = 96%]; ^1^H NMR (250 MHz, CDCl_3_) δ 5.47 (dq, *J*_2,1_ = 3.3 Hz, *J*_2,1′_ = 1.6 Hz, 1H, H-2), 4.22–4.02 (m, 1H, H-1), 2.06–1.81 (m, 2H, H-6), 1.86–1.57 (m, 2H, H-5), 1.67 (s, 3H, H-1′), 1.64–1.34 (m, 2H, H-4).

**(1*R*)-1,2,3,4-Tetrahydro-1-naphthalenol [(*R*)-15].** To a stirred solution of α-tetralone **10** (100 µL, 110 mg, 0.73 mmol) in a biphasic medium of CH_2_Cl_2_ (2.4 mL) and water (2.4 mL) and under a nitrogen flux, tetrabutylammonium chloride (61 mg, 0.22 mmol), sodium formate (149 mg, 2.19 mmol), and (*R,R*)-Noyori-I catalyst [(*R,R*)-**5**] (13.9 mg, 0.022 mmol) were sequentially added at room temperature. The reaction mixture was allowed to stir for 24 h. Then, CH_2_Cl_2_ (2 mL) and water (2 mL) were added, the two phases were separated, and the aqueous layer was extracted with CH_2_Cl_2_ (3 × 1 mL). The volatiles of the organic phase were removed under reduced pressure and the resulting residue was purified by column chromatography (CHCl_3_), yielding alcohol (*R*)-**15** (97 mg, 0.65 mmol, 87% yield) as a yellowish oil: CHPLC (Daicel Chiralpack OD-H): 94% ee; [α]_D_^20^ = −36.5 (*c* 0.98, CHCl_3_) [lit: [40] [α]_D_^20^ = −33.2 (*c* 1.0, CHCl_3_), ee = 99%]; ^1^H NMR (250 MHz, CDCl_3_) δ 7.43 (dd, *J*_8,7_ = 5.5 Hz, *J*_8,6_ = 3.5 Hz, 1H, H-8), 7.25–7.17 (m, 2H, H-6, H-7), 7.11 (dd, *J*_5,6_ = 5.5 Hz, *J*_5,7_ = 3.5 Hz 1H, H-5), 4.76 (t, *J*_1,2ax_ = *J*_1,2eq_ = 4.5 Hz, 1H, H-1), 2.94–2.62 (m, 2H, H-4), 2.05–1.84 (m, 4H, H-2, H-3); ^13^C NMR (100 MHz, CDCl_3_) δ 138.9 (C-8a), 137.3 (C-4a), 129.2/128.8 (C-5, C-8), 127.7/126.3 (C-6, C-7), 68.3 (C-1), 32.4 (C-2), 29.4 (C-4), 18.9 (C-3).

**(1*R*,5*R*,6*S*)-5-(Benzyloxy)bicyclo [4.1.0]heptan-2-one****(****18).** To a stirred solution of allylic alcohol (*R*)-**6** (314 mg, 2.01 mmol) in anhydrous CH_2_Cl_2_ (10 mL), Et_2_Zn (4 mL, 4.02 mmol, 1 M in hexane) at 0 ᵒC was added and the mixture was stirred for 5 min at this temperature. Then, a solution of ICH_2_Cl (600 µL, 8.30 mmol) in CH_2_Cl_2_ (3 mL) was added dropwise via syringe at 0 ᵒC, and the mixture was stirred overnight, allowing it to warm to room temperature. Then, a saturated aqueous NaHCO_3_ solution (10 mL) was added, and the aqueous layer was extracted with CH_2_Cl_2_ (3 × 10 mL). The organic layer was dried (Na_2_SO_4_), concentrated under reduced pressure (with vacuum controller) and purified by column chromatography (hexane-EtO_2_, 4:1) furnishing ketone **16** (198 mg, 1.57 mmol, 78% yield) as a colorless oil. Due to the volatility of compound **16**, the crude was used without further purification prior to the next step.

To a solution of crude of the hydroxyacetone **16** in CH_2_Cl_2_ (4 mL), Ag_2_O (560 mg, 2.41 mmol) and benzyl bromide (340 µL, 2.81 mmol) were added. The mixture was stirred at room temperature for 24 h, then it was filtered through a Celite^®^ pad and concentrated in vacuo. The crude was purified by column chromatography (hexanes-EtOAc, 6:1) to furnish **18** (270 mg, 1,24 mmol, 62% yield from (*R*)-**6**) as a pale oil.

**16:** [α]_D_^20^ = +78.5 (*c* 0.65, CHCl_3_) [lit: [41] [α]_D_^20^ = −80.7 (*c* 1.37, CHCl_3_), for its enantiomer, ee = 100%]; ^1^H NMR (400 MHz, CDCl_3_) δ 4.42 (dddd, *J*_5,4ax_ = 10.1 Hz, *J*_5,6_ = 5.2 Hz, *J*_5,4eq_ = 4.1 Hz, *J*_5,3ax_ = 0.8 Hz, 1H, H-5), 2.37 (ddd, *J*_gem_ = 18.4 Hz, *J*_3eq,4ax_ = 5.6 Hz, *J*_3eq,4eq_ = 3.6 Hz, 1H, H-3eq), 2.15 (dddd, *J*_gem_ = 18.4 Hz, *J*_3ax,4ax_ = 12.1 Hz, *J*_3ax,4eq_ = 6.6 Hz, *J*_3ax,5_ = 0.8 Hz, 1H, H-3ax), 1.98–1.83 (m, 3H, H-1, H-6, H-4eq), 1.63 (dddd, *J*_gem_ = 13.8 Hz, *J*_4ax,3ax_ = 12.1 Hz, *J*_4ax,5_ = 10.3 Hz, *J*_4ax,3eq_ = 5.6 Hz, 1H, H-4ax), 1.44 (q, *J*_gem_ = *J*_7endo,1_ = *J*_7endo,6_ = 5.4 Hz, 1H, H-7endo), 1.14 (ddd, *J*_7exo,1/6_ = 9.8 Hz, *J*_7exo,1/6_ = 7.7 Hz, *J*_gem_ = 5.4 Hz, 1H, H-7exo); ^13^C NMR (100 MHz, CDCl_3_) δ 207.5 (C-2), 65.3 (C-5), 34.8 (C-3), 26.8 (C-1), 26.7 (C-4), 23.3 (C-6), 8.0 (C-7); IR (ATR) ν 3359, 2919, 2850, 1659, 1345, 1066, 1041 (cm^−1^). HRMS (ESI+) Calcd. for [C_7_H_10_O_2_+Na]^+^ 149.0573, found: 149.0576.

**18**: [α]_D_^20^ = +62.6 (*c* 1.45, CHCl_3_); ^1^H NMR (400 MHz, CDCl_3_) δ 7.39–7.34 (m, 4H, H-Ar), 7.32–7.28 (m, 1H, H-Ar), 4.73 (d, *J*_gem_ = 11.9 Hz, 1H, C*H*_2_-Ph), 4.64 (d, *J*_gem_ = 11.9 Hz, 1H, C*H*_2_-Ph), 4.15 (dt, *J*_5,4ax_ = 9.3 Hz, *J*_5,4eq_ = *J*_5,6_ = 4.9 Hz, 1H, H-5), 2.42 (dt, *J*_gem_ = 17.7 Hz, *J*_3eq,4_ = 5.5 Hz, 1H, H-3eq), 2.12 (ddd, *J*_gem_ = 17.7 Hz, *J*_3ax,4x_ = 10.7 Hz, *J*_3ax,4eq_ = 6.5 Hz, 1H, H-3ax), 2.05–1.93 (m, 2H, H-4), 1.88–1.71 (m, 2H, H-1, H-6), 1.50 (q, *J*_gem_ = *J*_7endo,6_ = *J*_7endo,1_ = 5.4 Hz, 1H, H-7endo), 1.24 (td, *J*_7exo,6_ = *J*_7exo,1_ = 9.1 Hz, *J*_gem_ = 5.4 Hz, 1H, H-7exo); ^13^C NMR (100 MHz, CDCl_3_) δ 208.2 (C-2), 138.5 (C-Ar), 128.6/128.2/127.9/127.8 (C-Ar), 71.3 (C-5), 70.6 (*C*H_2_-Ph), 34.4 (C-3), 26.5 (C-4), 25.5 (C-1), 21.3 (C-6), 9.5 (C-7); IR (ATR) ν 3028, 2857, 1692 (C=O), 1342, 1075, 1028, 881, 631 cm^−1^. HRMS (ESI+) Calcd. for [C_14_H_16_O_2_+H]^+^ 217.1229, Found: 217.1231.

**[(1*S*’,2*R*’,5*R*’,6*S*’)-5′-(Benzyloxy)bicyclo [4.1.0]hept-2′-yl]methanol (21) and its (2′*S*)-diastereoisomer (22).** To a stirring solution of Ph_3_PCH_3_I (1.452 g, 3.59 mmol) in anhydrous THF (4 mL) at 0 °C, *t-*BuOK (402 mg, 3.58 mmol) was added, under nitrogen atmosphere, and the resulting yellow mixture was allowed to react for 1 h. Then, a solution of ketone **18** (155 mg, 0.71 mmol) in dry THF (1 mL) was added and the mixture was allowed to warm to room temperature and stirred for 3 h. Then, diethyl ether (10 mL) was added and the crude was filtered through a short pad of silica and Celite^®^, using additional diethyl ether as eluent. The volatiles were removed under vacuum to obtain an orange oil of alkene **19** that was used for the next step without further purification, as it isomerizes to the more stable endocyclic regioisomer **20** at room temperature.

The crude of alkene **19** was rapidly dissolved in anhydrous THF (7 mL) and 9-BBN (4.30 mL, 2.15 mmol, 0.5 M in THF) was added at −10 °C. The mixture was allowed to warm to room temperature and stirred overnight. Then, water (1.2 mL), NaOH (1.5 mL, 3 M in water), and H_2_O_2_ (1.4 mL, 30% in water) were added at 0 °C. After stirring for 15 min, the mixture was diluted with brine (15 mL) and CH_2_Cl_2_ (15 mL) and the aqueous phase was extracted with additional CH_2_Cl_2_ (2 × 10 mL). The organic layers were dried (Na_2_SO_4_), concentrated under reduced pressure, and purified by column chromatography (gradient hexane-EtOAc, 5:1 → 2:1) to provide a chromatographically inseparable mixture of alcohols **21** and **22** (158 mg, 0.68 mmol, 95% overall yield from **18**) in a ca. 2:1 diastereomeric ratio as a colorless oil. After repeated purification by column chromatography, enriched fractions were obtained and were analyzed by NMR.

**19**: ^1^H NMR (400 MHz, CDCl_3_) δ 7.46–7.25 (m, 5H, H-Ar), 4.90 (br s, 1H, H-1’), 4.79 (br s, 1H, H-1′), 4.74 (d, *J*_gem_ = 11.9 Hz, 1H, C*H*_2_-Ph), 4.56 (d, *J*_gem_ = 11.9 Hz, 1H, C*H*_2_-Ph), 4.05 (q, *J*_2,3ax_ = *J*_2,3eq_ = *J*_2,1_ = 5.9 Hz, 1H, H-2), 2.21 (dddt, *J*_gem_ = 14.9 Hz, *J*_4ax,3ax_ = 8.5 Hz, *J*_4ax,3eq_ = 4.4 Hz, *J*_4ax,1′_ = 1.4 Hz, 1H, H-4ax), 2.03 (dddt, *J*_gem_ = 14.9 Hz, *J*_4eq,3ax_ = 7.2 Hz, *J*_4eq,3eq_ = 4.3 Hz, *J*_4eq,1′_ = 1.4 Hz, 1H, H-4eq), 1.88–1.75 (m, 1H, H-6), 1.71–1.50 (m, 3H, H-1, H-3), 0.94–0.84 (m, 2H, H-7); ^13^C NMR (100 MHz, CDCl_3_) δ 146.3 (C-5), 139.1 (C-*ipso*), 128.4/127.7/127.5 (C-Ar), 108.1 (C-1′), 72.0 (C-2), 69.7 (*C*H_2_-Ph), 28.8 (C-4), 27.7 (C-3), 19.9 (C-1), 17.0 (C-6), 8.7 (C-7); IR (ATR) ν 3070, 2930, 1471, 1427, 1106, 806, 741 cm^−1^. HRMS (ESI+) Calcd. for [C_15_H_18_O+H]^+^ 215.1436, Found: 215.1465.

**20**: ^1^H NMR (400 MHz, CDCl_3_) δ 7.42–7.36 (m, 3H, H-Ar), 7.35–7.25 (m, 2H, H-Ar), 5.01 (dq, *J*_3,4eq_ = 5.3 Hz, *J*_3,4ax_ = *J*_3,1′_ = 1.6 Hz, 1H, H-3), 4.69 (d, *J*_gem_ = 12.0 Hz, 1H, C*H*_2_-Ph), 4.65 (d, *J*_gem_ = 12.0 Hz, 1H, C*H*_2_-Ph), 3.98 (ddd, *J*_5,4ax_ = 9.1 Hz, *J*_5,4eq_ = 6.8 Hz, *J*_5,6_ = 4.1 Hz, 1H, H-5), 2.32 (ddd, *J*_gem_ = 15.3 Hz, *J*_4eq,5_ = 6.8 Hz, *J*_4eq,3_ = 5.2 Hz, 1H, H-4eq), 1.89–1.75 (m, 4H, H-1′, H-4ax) 1.64–1.51 (m, 1H, H-6), 1.30–1.23 (m, 1H, H-1), 0.90–0.85 (m, 2H, H-7); ^13^C NMR (100 MHz, CDCl_3_): δ 139.6 (C-*ipso*), 136.7 (C-2), 128.8/128.2/127.9 (C-Ar), 114.3 (C-3), 73.4 (C-5), 70.6 (*C*H_2_-Ph), 28.0 (C-4), 23.5 (C-1′), 18.0/17.8 (C-1, C-6), 10.1 (C-7); IR (ATR) ν 3064, 2910, 2852, 1667, 1452, 1070, 734, 697 cm^−1^. HRMS (ESI+) Calcd. for [C_15_H_18_O+H]^+^ 215.1436, Found: 215.1434.

**21** and **22** mixture: ^1^H NMR (400 MHz, CDCl_3_) (*ca.* 84% *trans*-isomer **21**) δ 7.40–7.31 (m, 4H, H-Ar), 7.30–7.24 (m, 1H, H-Ar), 4.71 (d, *J*_gem_ = 12.0 Hz, 1H, C*H*_2_-Ph), 4.59 (d, *J*_gem_ = 12.0 Hz, 1H, C*H*_2_-Ph), 3.95 (dt, *J*_5′,4′ax_ = 10.0 Hz, *J*_5′,4′eq_ = *J*_5′,6′_ = 5.7 Hz, 1H, H-5′), 3.57 (d, *J*_1,2′_ = 6.6 Hz, 2H, H-1), 1.79 (dtd, *J*_gem_ = 13.3 Hz, *J*_4′eq,5′_ = *J*_4′eq,3′ax_ = 5.7 Hz, *J*_4′eq,3′eq_ = 2.4 Hz, 1H, H-4′eq), 1.70 (dddd, *J*_2′,3′ax_ = 8.5 Hz, *J*_2′,1_ = 6.6 Hz, *J*_2′,3′eq_ = 5.4 Hz, *J*_2′,1′_ = 1.9 Hz, 1H, H-2′), 1.55 (dtd, *J*_gem_ = 13.2 Hz, *J*_3′eq,4′ax_ = *J*_3′eq,2′_ = 5.4 Hz, *J*_3′eq,4′eq_ = 2.4 Hz, 1H, H-3′eq), 1.29–1.26 (m, 1H, H-6′), 1.04 (tdd, *J*_gem_ = *J*_4′ax,3′ax_ = 13.3 Hz, *J*_4′ax,5′_ = 10.0 Hz, *J*_4′ax,3′eq_ = 5.4 Hz, 1H, H-4′ax), 0.99–0.83 (m, 2H, H-1′, H-3′ax), 0.74 (td, *J*_7′exo,6′_ = *J*_7′exo,1′_ = 8.8 Hz, *J*_gem_ = 5.4 Hz, 1H, H-7′exo), 0.46 (q, *J*_gem_ = *J*_7′endo,6′_ = *J*_7′endo,1′_ = 5.4 Hz, 1H, H-7′endo); ^1^H NMR (400 MHz, CDCl_3_) (*ca.* 16% *cis*-isomer **22**, observable signals) δ 7.40–7.31 (m, 4H, H-Ar), 7.30–7.24 (m, 1H, H-Ar), 4.72 (d, *J*_gem_ = 11.8 Hz, 1H, C*H*_2_-Ph), 4.47 (d, *J*_gem_ = 11.8 Hz, 1H, C*H*_2_-Ph), 4.01 (ddd, *J*_5′,4′ax_ = 7.0 Hz, *J*_5′,4′eq_ = 5.1 Hz, *J*_5′,6′_ = 3.4 Hz, 1H, H-5′), 3.57 (m, 1H, H-1a), 3.49 (dd, *J*_gem_ = 10.5 Hz, *J*_1b,2′_ = 6.6 Hz, 1H, H-1b), 2.05 (dddd, *J*_2′,3′ax_ = 12.3 Hz, *J*_2′,1a_ = 10.2 Hz, *J*_2′,1b_ = 6.6 Hz, *J*_2′,1′_ = 5.4 Hz, 1H, H-2′), 1.66–1.57(m, 1H, H-4′a), 1.44–1.33 (m, 2H, H-3′a, H-4′b), 1.18 (tt, *J*_1′,6′_ = *J*_1′,7′exo_ = 8.5 Hz, *J*_1′,7′endo_ = *J*_1′,2′_ = 5.4 Hz, 1H, H-1′), 1.02–0.98 (m, 1H, H-3′b), 0.75 (m, 1H, H-6′), 0.60–0.51 (m, 2H, H-7′); ^13^C NMR (101 MHz, CDCl_3_) (*ca.* 84% *trans*-isomer **20**): δ 139.25 (C-*ipso*), 128.43 (C-*meta*), 127.82 (C-*para*), 127.49 (C-*orto*), 74.44 (C-5′), 69.62 (*C*H_2_-Ph), 67.98 (C-1), 38.10 (C-2′), 25.31 (C-4′), 25.23 (C-3′), 15.90 (C-1′), 14.88 (C-6′), 8.21 (C-7′); ^13^C NMR (100 MHz, CDCl3) (*ca.* 16% *cis*-isomer **21**, observable signals) δ 139.22 (C-*ipso*), 128.43 (C-*meta*), 127.80 (C-*para*), 127.44 (C-*orto*), 70.87 (C-5′), 69.57 (*C*H_2_-Ph) 67.78 (C-1), 35.64 (C-2′), 28.43 (C-4′), 18.93 (C-3′), 14.19 (C-1′), 13.37 (C-6′), 1.98 (C-7′). HRMS (ESI+) Calcd. for [C_15_H_20_O_2_+H]^+^ 233.1542, Found: 233.1537.

**[(1′*S*,2′*R*,5′*R*,6′*S*)-5′-Hydroxybicyclo [4.1.0]hept-2′-yl]methyl benzoate (23) and its (2′*S*)-diastereoisomer (24).** To a stirred solution of a 2:1 mixture of **21** and **22** (158 mg, 0.68 mmol) in dry CH_2_Cl_2_ (7 mL) at 0 °C, anhydrous Et_3_N (100 µL, 0.71 mmol) and benzoyl chloride (91 µL, 0.78 mmol) were sequentially added under argon atmosphere. The mixture was subsequently allowed to attain room temperature overnight. Then, the solution was treated with HCl 10% solution (7 mL) and CH_2_Cl_2_ (7 mL), the two phases were separated, and the aqueous phase was extracted with CH_2_Cl_2_ (3 × 7 mL). The organic layers were washed with brine (30 mL), dried (Na_2_SO_4_), and concentrated under reduced pressure to obtain a colorless oil that was directly used for the next step without further purification. Accordingly, the crude was dissolved in EtOH (7 mL) and hydrogenated in the presence of Pd/C (23 mg, 10% wt.) at 2 atm overnight. The mixture was filtered through a short pad of Celite^®^ and washed with additional EtOH. The solvent was evaporated under reduced pressure and purified by column chromatography (hexanes-EtOAc, 10:1 → 5:1) to afford alcohol **23** (104 mg, 0.42 mmol, 62% yield) as a colorless syrup and its isomer **24** (49 mg, 0.20 mmol, 29% yield) as a colorless syrup.

**23**: [α]_D_^20^ = +48.7 (*c* 1,02, CHCl_3_); ^1^H NMR (400 MHz, CDCl_3_) δ 8.05 (d, *J_orto,meta_* = 7.6 Hz, 2H, H-*orto*), 7.61–7.51 (m, 1H, H-*para*), 7.44 (t, *J_meta,orto_* = *J_meta,para_* = 7.6 Hz, 2H, H-*meta*), 4.28 (dd, *J*_1,2′_ = 6.8 Hz, *J*_1,1′_ = 2.0 Hz, 2H, H-1), 4.20 (dt, *J*_5′,4′ax_ = 9.5 Hz, *J*_5′,4′eq_ = *J*_5′,6′_ = 5.7 Hz, 1H, H-5′), 2.0 (dddd, *J*_2′,3′ax_ = 13.8 Hz, *J*_2′,1_ = 6.8 Hz, *J*_2′,3′eq_ = 4.4 Hz, *J*_2′,1′_ = 1.9 Hz, 1H, H-2′), 1.86–1.72 (m, 1H, H-4′), 1.70–1.61 (m, 2H, H-3′), 1.33 (tt, *J*_6′,7′endo_ = *J*_6′,7′exo_ = 8.8 Hz, *J*_6′,5′_ = *J*_6′,1′_ = 5.7 Hz, 1H, H-6′), 1.13–0.87 (m, 2H, H-1′, H-4′), 0.71 (td, *J*_7′exo,1′_ = *J*_7′exo,6′_ = 8.8 Hz, *J*_gem_ = 4.9 Hz, 1H, H-7′exo), 0.40 (q, *J*_7′endo,1′_ = *J*_7′endo,6′_ = *J*_gem_ = 5.3 Hz, 1H, H-7′endo); ^13^C NMR (100 MHz, CDCl_3_) δ 166.8 (*C*=O), 133.1 (C-*para*), 130.4 (C-*ipso*), 129.7 (C-*orto*), 128.5 (C-*meta*), 69.4 (C-1), 68.0 (C-5′), 34.5 (C-2′), 28.0 (C-4′), 26.0 (C-3′), 18.1 (C-6′), 16.2 (C-1′), 6 (C-7′).

**24:** [α]_D_^20^ = −12.3 (*c* 1.16, CHCl_3_); ^1^H NMR (400 MHz, CDCl_3_) δ 8.05 (d, *J_orto,meta_* = 7.2 Hz, 2H, H-*orto*), 7.56 (t, *J_para,meta_* = 7.4 Hz, 1H, H-*para*), 7.44 (t, *J_meta,orto_* = *J_meta,para_* = 7.6 Hz, 2H, H-*meta*), 4.34 (dt, *J*_5′,4′ax_ = 7.3 Hz, *J*_5′,4′eq_ = *J*_5′,6′_ = 4.7 Hz, 1H, H-5′), 4.29 (dd, *J*_gem_ = 10.7 Hz, *J*_1a,2′_ = 6.9 Hz, 1H, H-1a), 4.18 (dd, *J*_gem_ = 10.7 Hz, *J*_1b,2′_ = 7.3 Hz, 1H, H-1b), 2.36 (dq, *J*_2′,3′ax_ = 12.6 Hz, *J*_2′,1a_ = *J*_2′,1b_ = *J*_2′,3′eq_ = 6.7 Hz, 1H, H-2′), 1.51–1.37 (m, 4H, H-6′, 2H-4′, H-3′), 1.27 (tt, *J*_1′,2′_ = *J*_1′,7′exo_ = 8.5 Hz, *J*_1′,7′endo_ = *J*_1′,6′_ = 5.7 Hz, 1H, H-1′), 1.23–1.13 (m, 1H, H-3′), 0.57 (q, *J*_gem_ = *J*_7′endo,6′_ = *J*_7′endo,1′_ = 5.3 Hz, 1H, H-7′endo), 0.46 (td, *J*_7′exo,1′_ = *J*_7′exo,6′_ = 8.9 Hz, *J*_gem_ = 5.1 Hz, 1H, H-7′exo); ^13^C NMR (100 MHz, CDCl_3_) δ 166.8 (*C*=O), 133.0 (C-*para*), 130.6 (C-*ipso*), 129.7 (C-*orto*), 128.5 (C-*meta*), 68.9 (C-1), 64.7 (C-5′), 31.7 (C-2′), 29.9 (C-4′), 19.4 (C-3′), 17.1 (C-1′), 14.7 (C-6′), 1.7 (C-7′).

**(1*S*,2*S*,5*R*,6*R*)-5-[(Benzoyloxy)methyl)]bicyclo [4.1.0]heptan-2-amonium chloride (25).** To a stirred solution of Ph_3_P (170 mg, 0.65 mmol) in dry toluene (4.5 mL), DBAD (150 mg, 0.65 mmol) was slowly added under argon atmosphere and the mixture was stirred for 45 min at 0 °C. After 15 min, a white suspension appeared. Then, diphenylphosphoryl azide (DPPA, 100 µL, 0.45 mmol) and a solution of **23** (105 mg, 0.43 mmol) in dry toluene (1 mL) were sequentially added. The mixture was allowed to warm to room temperature and stirred overnight. Then, the solvent was removed, and the crude was purified by column chromatography (hexanes-EtOAc, 20:1 to 15:1) to obtain a crude that was dissolved in EtOAc (2 mL) and hydrogenated in the presence of Pd/C (15 mg, 10% wt.) at 2 atm for 24 h. Then, the mixture was filtered through a short pad of Celite^®^ and washed with additional EtOAc. The solvent was evaporated under reduced pressure and the crude was treated with 2 M HCl in Et_2_O (2 mL, 4 mmol) at 0 °C, stirred for 2 h, and filtered to furnish the ammonium salt **25** (101 mg, 0.36 mmol, 84% yield) as a brown solid: Mp 130–132 °C (from EtO_2_); [α]_D_^20^ = +14.2 (*c* 1.0, CHCl_3_); ^1^H NMR (400 MHz, CDCl_3_) δ 8.69 (s, 3H, N*H*_3_^+^), 8.08–7.97 (m, 2H, H-Ar), 7.56–7.50 (m, 1H, H-Ar), 7.44–7.36 (m, 2H, H-Ar), 4.45 (d, *J*_1′,5_ = 7.7 Hz, 2H, H-1′), 3.84 (br s, 1H, H-2), 2.15–1.89 (m, 2H, H-5, H-4), 1.60–1.46 (m, 3H, H-3, H-4), 1.31–1.22 (m, 1H, H-1), 1.12–1.02 (m, 1H, H-6), 0.91 (td, *J*_7exo,1_ = *J*_7exo,6_ = 9.1 Hz, *J*_gem_ = 5.1 Hz, 1H, H-7exo), 0.19 (q, *J*_gem_ = *J*_7endo,1_ = *J*_7endo,6_ = 5.5 Hz, 1H, H-7endo); ^13^C NMR (100 MHz, CDCl_3_) δ 166.6 (*C*=O), 133.1 (C-*para*), 130.4 (C-*ipso*), 129.7 (C-*orto*), 128.5 (C-*meta*), 69.0 (C-1′), 47.1 (C-2), 33.9 (C-5), 23.2 (C-4), 19.4 (C-3), 14.0 (C-6), 12.5 (C-1), 10.4 (C-7); IR (ATR) ν 3404, 2929, 1712, 1273, 1113, 713 cm^−1^. HRMS (ESI+) Calcd. for [C_15_H_18_NO_2_]^+^ 244.1338, Found: 244.1334.

**(*E*)-3-Ethoxyacryloyl isocyanate (26).** Vinyl ether (5.75 mL, 60 mmol) was added dropwise to oxalyl chloride (7.60 mL, 90 mmol) at 0 °C. The reaction mixture was maintained for 2 h at 0 °C and then warmed to room temperature overnight. Excess oxalyl chloride was distilled off and the black residue was heated at 120 °C for 30 min. Then, the residue was purified by vacuum distillation, using a short Vigreux column, to obtain (2*E*)-3-ethoxyacryloyl chloride (4.30 g, 31.97 mmol, 53% yield) as a colorless liquid.

Silver cyanate (90 mg, 0.60 mmol), previously dried over phosphorus pentoxide at 80 °C for 3 h, in dry benzene (2 mL), was heated to reflux for 30 min and a solution of (2*E*)-3-ethoxyacryloyl chloride (45 mg, 0.31 mmol) in dry benzene (0.8 mL) was then added dropwise. The mixture was stirred for 30 min before allowing the solid to settle. The supernatant, which is a solution of the isocyanate **25**, was then decanted and used directly in the next reaction.

**[(1′*R*,2′*R*,5′*S*,6′*S*)-5′-(2″,4″-dioxo-3″,4″-dihydropyrimidin-1″(2*H*)-yl] bicyclo [4.1.0]heptan-2′-yl)methyl benzoate (27)**. The ammonium chloride **25** (43 mg, 0.153 mmol) was dissolved in dry DMF (1.8 mL), and Et_3_N (22 µL, 0.157 mmol) was added. The mixture was cooled to −20 °C and a freshly prepared solution of the isocyanate **26** was added slowly to avoid an increase in the temperature. The reaction mixture was stirred overnight at room temperature. The solvent was evaporated in vacuo, and then water (2 mL) was added. The residue was extracted with EtOAc (2 × 2 mL), washed with brine (2 mL), dried (Na_2_SO_4_), filtered, and evaporated under reduced pressure. The residue was dissolved in MeOH (0.40 mL), H_2_SO_4_ (1 M, 0.62 mL) was added, and the mixture was heated to reflux for 3 h. Then, the mixture was concentrated under reduced pressure and purified by column chromatography (CH2Cl2 100% to CH_2_Cl_2_-MeOH, 20:1) to provide **27** (25 mg, 0.073 mmol, 48% yield) as a yellowish oil: ^1^H NMR (400 MHz, MeOH-*d*_4_) δ 8.05 (d, *J_orto,meta_* = 7.6 Hz, 2H, H-*orto*), 7.96 (d, *J*_6″,5″_ = 8.0 Hz, 1H, H-6″), 7.64 (d, *J_para,meta_* = 7.6 Hz, 1H, H-*para*), 7.52 (t, *J*_meta,orto_ = *J*_meta,para_ = 7.6 Hz, 2H, H-*meta*), 5.46 (d, *J*_5″,6″_ = 8.0 Hz, 1H, H-5″), 4.83 (dt, *J*_5′,4′ax_ = 4.3 Hz, *J*_5′,4eq_ = *J*_5′,6′_ = 2.3 Hz, 1H, H-5′), 4.45 (d, *J*_1,2′_ = 6.0 Hz, 2H, H-1), 2.21 (dq, *J*_2′,3′ax_ = 11.9 Hz, *J*_2′,3′eq_ = *J*_2′1_ = 6.0 Hz, 1H, H-2′), 1.78–1.68 (m, 1H, H-4′eq), 1.54 (tt, *J*_gem_ = *J*_4′ax,3′ax_ = 14.7 Hz, *J*_4′ax,5′_ = *J*_4′ax,3′eq_ = 4.3 Hz, 1H, H-4′ax), 1.49–1.40 (m, 1H, H-3′), 1.32–1.21 (m, 2H, H-3′, H-6′), 1.12–1.04 (m, 1H, H-1′), 0.97 (td, *J*_7exo,1_ = *J*_7exo,6_ = 9.4 Hz, *J*_gem_ = 5.0 Hz, 1H, H-7′exo), 0.39 (q, *J*_gem_ = *J*_7endo,1_ = *J*_7endo,6_ = 5.3 Hz, 1H, H-7′endo); ^13^C NMR (101 MHz, MeOH-*d*_4_): δ 168.1 (C=O), 166.3 (C-4″), 152.7 (C-2″), 145.1 (C-6″), 134.4 (C-*para*), 131.5 (C-*ipso*), 130.5 (C-*orto*), 129.7 (C-*meta*), 101.6 (C-5″), 69.9 (C-1), 52.5 (C-5′), 34.7 (C-2′), 24.5 (C-4′), 20.1 (C-3′), 15.3 (C-6′), 13.7 (C-1′), 10.4 (C-7′). HRMS (ESI+) Calcd. for [C_19_H_20_N_2_O_4_+Na]^+^ 363.1321, Found: 363.1312.

**4-Amino-1-[(1′*S*,2′*S*,5′*R*,6′*R*)-5′-(hydroxymethyl)bicyclo [4.1.0]hept-2′-yl]pyrimidin-2(1*H*)-one (2e).** A solution of TsCl (31 mg, 0.16 mmol) in dry CH_3_CN (104 µL) was added to a mixture of **27** (27 mg, 0.08 mmol), Et_3_N (22 µL, 0.16 mmol), and *N*-methylpiperidine (12 µL, 0.1 mmol) in dry CH_3_CN (135 µL) at 0 °C, and the reaction mixture was stirred for 3 h. Then, 30% NH_4_OH was added at 0 °C, and the reaction solution was stirred at room temperature overnight. The mixture was diluted with water (1 mL) and EtOAc (1 mL) and the aqueous phase was extracted with additional CH_2_Cl_2_ (2 × 1 mL). The organic layers were dried (Na_2_SO_4_), concentrated under reduced pressure, and purified by column chromatography (CH_2_Cl_2_-EtOAc, 10:2 → CH_2_Cl_2_-MeOH 15:1) to provide the protected cytosine analogue as a yellowish oil that was dissolved in a 33% solution of methylamine in EtOH (17 mL) and stirred for 24 h. Then, the mixture was concentrated under reduced pressure and purified by column chromatography (CH_2_Cl_2_-MeOH, 20:1) to provide cytosine nucleoside analogue **2e** (9 mg, 38 μmol, 50% yield) as a yellowish solid: **2e:** [α]_D_^20^ = +36.2 (*c* 0.4, MeOH-d_4_); ^1^H NMR (400 MHz, MeOH-*d*_4_) δ 8.04 (d, *J*_6,5_ = 7.4 Hz, 1H, H-6), 5.90 (d, *J*_5,6_ = 7.4 Hz, 1H, H-5), 4.91–4.87 (m, 1H, H-2′) 3.64 (dd, *J*_gem_ = 10.7 Hz, *J*_1″a,5′_ = 5.9 Hz, 1H, H-1″a), 3.59 (dd, *J*_gem_ = 10.7 Hz, *J*_1″a,5′_ = 5.9 Hz, 1H, H-1″b), 1.85 (dq, *J*_5′,4′ax_ = 11.4 Hz, *J*_5′4′eq_ = *J*_5′,1″a_ = *J*_5′,1″b_ = 5.9 Hz, 1H, H-5′), 1.68 (dt, *J*_gem_ = 13.7 Hz, *J*_3′eq,4′ax_ = *J*_3′eq,5′_ = 4.4 Hz, 1H, H-3′eq), 1.48 (tt, *J*_gem_ = *J*_3′ax,4′ax_ = 13.7 Hz, *J*_3′ax, 4′eq_ = *J*_3′ax,5′_ = 3.6 Hz, 1H, H-3′ax), 1.34–1.26 (m, 2H, H-1′, H-4′a), 1.19–1.07 (m, 1H, H-4′b), 1.00 (td, *J*_6′,7′exo_ = *J*_6′,1′_ = 9.4 Hz, *J*_6′,7′endo_ = 5.2 Hz, 1H, H-6′), 0.89 (td, *J*_7exo,1_ = *J*_7exo,6_ = 9.4, *J*_gem_ = 5.2 Hz, 1H, H-7′exo), 0.30 (q, *J*_gem_ = *J*_7endo,1_ = *J*_7endo,6_ = 5.2 Hz, 1H, H-7′endo); ^13^C NMR (100 MHz, MeOH-*d*_4_): δ 167.3 (C-4), 159.0 (C-2), 145.6 (C-6), 95.0 (C-5), 67.6 (C-1″), 52.8 (C-2′), 37.5 (C-5′), 24.6 (C-3′), 19.6 (C-4′), 15.7 (C-1′), 13.81 (C-6′), 10.26 (C-7′). HRMS (ESI+) Calcd. for [C_12_H_17_N_3_O_2_+H]^+^ 236.1399, Found: 236.1394.

## 4. Conclusions

In this article, an easy route to enantiomerically pure 4-hydroxy-2-cyclohexanone derivatives **1a**,**b**, which are commonly used as chiral building blocks, was identified on the basis of ATH with bifunctional ruthenium catalysts. This approach provided convenient access to both enantiomers from the same precursor by selecting either the (*S*,*S*)- or (*R*,*R*)-**5** catalyst. In addition, a stereoselective route to the cytosine analogue, built on a bicyclo [4.1.0]heptane scaffold, was finely tuned, starting from cyclohexanol (*R*)-**6**. Finally, the kinase activity assays of compounds **2a**–**e** showed that compounds **2b** and **2d** display affinity for HSV-1 TK but not for human TK-1, thus validating our previous molecular modeling study and pointing to their selectivity for herpetic TKs but not cellular TKs.

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
