# Peer review of "Enantiocontrolled Preparation of ϒ-Substituted Cyclohexenones: Synthesis and Kinase Activity Assays of Cyclopropyl-Fused Cyclohexane Nucleosides"

_ijms, 2022, doi:10.3390/ijms23179704_

Round 1
Reviewer 1 Report
The authors described synthesis of cyclohexane nucleoside with high enantioselectivity starting from cyclohexenone via ATH reaction in the presence of a chiral organoruthenium catalyst. The resulting nucleoside 2e alone with previously prepared 2a-2d were tested with nucleoside kinase activity assays, and some of them exhibited good selectivity for HSV-1 versus human TK. Due to the importance of the chiral cycloalkane nucleosides as potential antiherpetic agents, this work herein is valuable and interesting. The manuscript was well written in a scientific manner, which could be suitable to be published in the International Journal of Molecular Sciences.
Before accepting for publication, I would suggest providing more examples of cyclic ketone substrates for asymmetric hydrogenation in Table 1, which would be useful to expand the application for the ATH reaction.
Author Response
Reviewer 1
“Before accepting for publication, I would suggest providing more examples of cyclic ketone substrates for asymmetric hydrogenation in Table 1, which would be useful to expand the application for the ATH reaction.”
The work describe in the manuscript is mainly focused on the preparation of enantiopure cyclohexenones and their use as precursors in the preparation of more elaborated compounds.
The ATH reaction was also studied on other cycloalkenones of five- and seven-membered rings. However, the results obtained were worse than those obtained with the cyclohexenones. As suggested by the reviewer, we have included a sentence in the new manuscript file (page 3, lines 106-108) mentioning that ”The ATH reaction has also been studied on cycloalkenones with five- and seven-membered rings getting lower values of enantio- and chemoselectivity (Supporting Information)”
All the information about this study has been included in the new SI (Table S1: Asymmetric Transfer Hydrogenation (ATH) study on other cycloalkenones of five and seven-membered rings; Experimental procedures, 1H NMR spectra, CHPLC and references).

Reviewer 2 Report
The manuscript reports efforts at the synthesis of chiral cyclopropys-fused cyclohexane nucleosides and a short study of the thymidine kinase activity of the products. The preparative efforts are aimed at developing new agents against human herpes virus. While the preparative work has been done and described in a professional manner (see also Supplementary Material), the pharmacological goals brought only modest results. The preparative experiments are nice examples of the use of Noyori-type chiral Ru catalysts.
The paper can be published, but some considerations are adviced to the Authors (and the Editor):
(1) The title is too long. A more concentrated title appears to the Reviewer better. E. G. Chirality induction in the synthesis of substituted cyclohexanones... or something similar.
(2) A mother-languague control of the English would be advisable. Some sentences are formulated without the required clarity, e. g. rows 33/35.
(3) For biological testing the products the Authors used Human Embrio cell lines. A declaration of respecting the ehical requirements would be necessary. The Authors are adviced to study (and eventually cite) the following publication:
Geraghty, R. J., Capes-Davis, A., Davis, J. M. + 9 Authors:
Guidelines for the use of cell lines in biomedical research.
British Journal of Cancer, 2014, 111, 1021-1046.
Author Response
Reviewer 2
(1) The title is too long. A more concentrated title appears to the Reviewer better. E. G. Chirality induction in the synthesis of substituted cyclohexanones... or something similar.
We appreciate the reviewer’s suggestion, but we sincerely believe that the current title perfectly reflects what the manuscript is focused on; enantioselective synthesis of ϒ-substituted cyclohexenones and the preparation of cyclohexane nucleoside analogues.
(2) A mother-languague control of the English would be advisable. Some sentences are formulated without the required clarity, e. g. rows 33/35.
The manuscript has been thoroughly revised to find spelling mistakes and have been corrected.
(3) For biological testing the products the Authors used Human Embrio cell lines. A declaration of respecting the ehical requirements would be necessary. The Authors are adviced to study (and eventually cite) the following publication: British Journal of Cancer, 2014, 111, 1021-1046.
This cell line is used all over the world for this purpose for many years. This is the first time we got such request. However, as suggested by the reviewer we have included the following declaration along with the proposed reference in the new SI.
“Adherent human embryonic lung fibroblast cell cultures ( HEL-299) were used ( ATCC CCL 137). These cells were obtained from a male in 1965. The cells were used for research purposes only, and cultivated in a containment level 2 in accordance to the Advisory Committee on Dangerous Pathogens (ACDP) Guidelines.[3]”
[3] Geraghty, R. J.; Capes-Davis, A.; Davis, J. M.; Downward, J. Freshney, R. I.; Knezevic, I.; Lovell-Badge, R.; Masters,J. R. W.; Meredith, J.; Stacey, G. N.; Thraves, P.; Vias, M. Guidelines for the Use of Cell Lines in Biomedical Research. Br. J. Cancer. 2014, 111, 1021–1046.
